# A Safety Warning Model Based on IAHA-SVM for Coal Mine Environment

**DOI:** 10.3390/s23146614

**Published:** 2023-07-22

**Authors:** Zhen Li, Feng Feng

**Affiliations:** School of Information Engineering, Ningxia University, Yinchuan 750021, China; lliknz@163.com

**Keywords:** coal mine safety, artificial hummingbird algorithm, Tent chaotic mapping, Levy flight, simplex method, support vector machine

## Abstract

Coal is an important resource that is closely related to people’s lives and plays an irreplaceable role. However, coal mine safety accidents occur from time to time in the process of working underground. Therefore, this paper proposes a coal mine environmental safety early warning model to detect abnormalities and ensure worker safety in a timely manner by assessing the underground climate environment. In this paper, support vector machine (SVM) parameters are optimized using an improved artificial hummingbird algorithm (IAHA), and its safety level is classified by combining various environmental parameters. To address the problems of insufficient global exploration capability and slow convergence of the artificial hummingbird algorithm during iterations, a strategy incorporating Tent chaos mapping and backward learning is used to initialize the population, a Levy flight strategy is introduced to improve the search capability during the guided foraging phase, and a simplex method is introduced to replace the worst value before the end of each iteration of the algorithm. The IAHA-SVM safety warning model is established using the improved algorithm to classify and predict the safety of the coal mine environment as one of four classes. Finally, the performance of the IAHA algorithm and the IAHA-SVM model are simulated separately. The simulation results show that the convergence speed and the search accuracy of the IAHA algorithm are improved and that the performance of the IAHA-SVM model is significantly improved.

## 1. Introduction

Coal, as a precious resource on Earth, has played a significant role in the economic development of many countries, leading to increased surface and underground mining activities in countries with abundant coal reserves [1]. However, most coal mines are prone to safety hazards due to the unpredictable nature of coal seams and the inherent risks associated with underground mining [2]. Various types of mining disasters have occurred, such as roof accidents, gas explosions, fires, flooding, and gas poisoning [3]. These disasters not only pose a serious threat to the safety of miners but also result in substantial property damage for coal mining enterprises. The essence of safety accidents is closely related to the mining environment. For example, methane $\left(CH_{4}\right)$ is the primary component of gas, and timely monitoring of methane concentration in the mining environment, coupled with appropriate protective measures, can greatly reduce the risk of gas explosions. In the event of a fire, the temperature rises rapidly, but real-time monitoring of temperature changes in the mine and prompt firefighting measures can help reduce casualties. Toxic gases in underground mines, including CO, $NH_{3}$, $H_{2}S$, and $SO_{2}$, can cause harm to workers’ health. Timely monitoring of toxic gas concentrations in the current area serves as a guarantee for miners’ well-being. Therefore, strengthening coal mine environmental monitoring and early warning systems is an effective means to reduce accidents and holds great significance for ensuring coal mine safety and production.

Figure 1 illustrates a commonly used framework for underground environmental monitoring systems, consisting of two main parts: surface (above ground) and underground components. The surface part primarily comprises a switch and user terminals, while the underground part consists of monitoring substations, ring network switches, power supplies, and sensors. Various sensors are employed to monitor the actual underground environment, and the data are transmitted in real time to the surface, enabling users to observe the underground conditions at any given time. With the continuous advancement of Internet of Things (IoT) technology, wireless sensor networks (WSNs) have become vital tools for underground environmental monitoring. Many experts utilize WSNs to collect underground environmental data for further research [4,5,6,7]. The data used in this study were also collected in real-time through WSNs.

With the development of computer technology, machine learning has gradually become integrated into our daily lives [8], and its applications in coal mine safety are extensive. Kumari et al. proposed a deep learning model combining uniform manifold approximation and projection (UMAP) with long short-term memory (LSTM) in their paper [9]. They used this model to predict the fire status in sealed areas of coal mines, aiding in the early implementation of fire control strategies to prevent further losses caused by mine fires. Slezak et al., in their publication [10], combined machine learning techniques with sensor technology to develop a decision support system, which they applied to predict the risk level of coal mine methane. The system demonstrated good predictive accuracy. In another study by Jo et al. [11], the authors proposed an underground mine air quality prediction system based on Azure machine learning. This system allows for the rapid assessment and prediction of air quality in coal mines, providing a safety guarantee for the underground environment. These examples demonstrate the necessity of applying machine learning to analyze various coal mine safety data for risk assessment, ensuring efficiency and accuracy.

Intelligent optimization algorithms are an important research area within machine learning. These algorithms based on heuristic principles can overcome the uncertainty and complexity of problems during the optimization process, making them suitable for various types of optimization problems [12]. Some well-known algorithms in this field include particle swarm optimization (PSO) [13], grey wolf optimizer (GWO) [14], and the whale optimization algorithm (WOA) [15], among others. In recent years, researchers worldwide have conducted in-depth studies on these algorithms and applied them to complex optimization problems.The artificial hummingbird algorithm (AHA) is an intelligent optimization algorithm proposed by Zhao et al. in 2022 [16]. It simulates the unique flight skills and intelligent foraging strategies of hummingbirds in nature for optimization purposes. Compared to other intelligent optimization algorithms, AHA has the advantages of having fewer parameters, a stable convergence speed, strong optimization capabilities, and high implementation efficiency. However, when solving complex problems, AHA may encounter issues such as becoming trapped in local optima, insufficient global exploration capability, and slow convergence speed. To address these limitations, experts have proposed various methods to improve the algorithm, making it more suitable for a wide range of optimization problems.

Support vector machine (SVM) is a type of generalized linear classifier used for binary classification in supervised learning [17]. Compared to other classification algorithms, SVM offers several advantages, including simplicity, robustness, and generality. As a result, many researchers have applied SVM to various application domains for classification tasks. SVM relies on two important parameters: the kernel function parameter (*C*) and the penalty factor (*g*), which determine the performance of SVM. However, these parameters are often selected manually, and choosing inappropriate values can lead to overfitting or underfitting. To ensure the proper selection of the kernel function parameter (*C*) and the penalty factor (*g*), researchers frequently employ intelligent optimization algorithms to optimize these parameters [18,19,20].

Inspired by the aforementioned research, we propose using an intelligent optimization algorithm combined with SVM to construct a coal mine environmental safety warning model. To improve the model’s performance, we employ an enhanced version of the artificial hummingbird algorithm (AHA) to optimize the kernel function parameter (*C*) and the penalty factor (*g*). In this study, we introduce an improved version called IAHA, which incorporates the following modifications: (1) initialization of the population using a fusion of the Tent chaotic map and reverse learning strategy, (2) introduction of the Levy flight strategy during the food-guiding phase, and (3) incorporation of the simplex method at the end of each iteration to replace the worst solution. We evaluate the effectiveness and accuracy of the improved IAHA algorithm by conducting simulation experiments on eight benchmark test functions. We compare IAHA with four basic algorithms, three single-stage improved AHA algorithms, and two existing improved artificial hummingbird algorithms. The results demonstrate the effectiveness and accuracy of the proposed improvements. Furthermore, we combine IAHA with SVM to predict the safety level of coal mine environments. We conduct simulation experiments to evaluate the performance of the improved model in this context.

The organization of this paper is as follows:

In Section 2, we provide an introduction to the basic theories of the artificial hummingbird algorithm and support vector machine.

In Section 3, we propose an improved artificial hummingbird algorithm, namely IAHA, and analyze its performance.

In Section 4, we utilize the improved algorithm to optimize the parameters of SVM and construct the IAHA-SVM coal mine environmental safety warning model. We compare the performance of this model with that of the AHA-SVM model optimized by the basic AHA algorithm.

In Section 5, we summarize the findings of this study and provide an outlook for future work.

By following this organizational structure, we aim to present a comprehensive study on the application of the IAHA algorithm combined with SVM in the prediction of coal mine environmental safety.

## 2. Related Work

In this section, we provide an introduction to the fundamental theories of the artificial hummingbird algorithm (AHA) and support vector machine (SVM). This foundation is crucial for the subsequent discussions on algorithm improvements and model construction.

### 2.1. Artificial Hummingbird Algorithm

AHA is an algorithm that simulates the flight skills and foraging behavior of hummingbirds. It consists of four main phases: initialization, food guiding, area foraging, and migration foraging. In the AHA algorithm, a memory function is implemented by introducing an access table. This table records the quality level of different food sources chosen by hummingbirds. The access level is determined by the time interval since the last visit to the food source. A longer time interval corresponds to a higher access level, and hummingbirds prioritize higher-access-level food sources. If two food source levels are the same, hummingbirds choose the one with the highest nectar replenishment rate, which is determined by the current fitness value. By incorporating these mechanisms, the AHA algorithm allows hummingbirds (representing solutions in the algorithm) to efficiently search for optimal solutions in the search space. The memory function and selection criteria based on the nectar replenishment rate help guide the exploration and exploitation of the search space, enabling the algorithm to find high-quality solutions.

The three flight skills of the hummingbird are axial flight, diagonal flight, and omnidirectional flight, which are used to control the direction vectors in dimensional space. Axial flight is described as follows:(1)$$D^{\left(i\right)}=\left\{\begin{array}{c} 1,\;i=randi([1,\;d])\\ 0,\;else \end{array}\right.,\;i=1,\;\ldots ,\;d$$

Diagonal flight is described as follows:(2)$$\begin{array}{c} D^{\left(i\right)}=\left\{\begin{array}{c} 1,\;i=p(j),\;j\in [1,k],\;i=1,\ldots ,d\\ 0,\;else \end{array}\right.\\ P=randperm\left(k\right),\;k\in \left[2,\left\lceil r_{1}\left(d-2\right)\right\rceil +1\right] \end{array}$$

Omnidirectional flight is defined as follows:(3)$$D^{\left(i\right)}=1,\;i=1,\;\ldots ,\;d$$
where $D^{\left(i\right)}$ indicates flight skills, randi([1,d]) indicates the generation of random integers from 1 to *d*, randperm(k) denotes a random integer arrangement from 1 to *k*, and $r_{1}$ denotes a random number in the range of [0, 1].

#### 2.1.1. Initialization

*n* hummingbirds are placed on each food source, and the food source locations are initialized as follows:(4)$$x_{i}=V_{L}+r\times \left(V_{U}-V_{L}\right),\;i=1,\;2,\;3\ldots n$$
where $x_{i}$ denotes the *i*th food source location; $V_{U}$ and $V_{L}$ are the upper and lower limits of the search space, respectively; *r* is a random number in the range of [0, 1]; and *n* is the population size.

The food source access table is initialized as follows:(5)$$S_{i,\;j}=\left\{\begin{array}{c} 0,\;i\neq j\\ \mathrm{null},\;i=j \end{array}\right.,\;i=1,\;\ldots ,\;n,\;j=1,\;\ldots ,\;n$$
where i=j indicates that the hummingbird is foraging at a specific food source, and i≠j indicates that the *j*th food source is currently visited by the *i*th hummingbird.

#### 2.1.2. Guided Foraging

When rand<0.5, hummingbirds follow their current flight skill to guide foraging by visiting the target food source, thus obtaining the candidate food source. The candidate food source location is updated as follows:(6)$$v_{i}\left(t+1\right)=x_{i,tar}\left(t\right)+a\times D\times \left[x_{i}\left(t\right)-x_{i,tar}\left(t\right)\right]$$
where $v_{i}\left(t+1\right)$ denotes the position of the *i*th candidate food source after t+1 iterations, $x_{i}\left(t\right)$ denotes the position of the *i*th candidate food source in the *t*th iteration, $x_{i,tar}\left(t\right)$ denotes the position of the *i*th hummingbird to visit the target food source, and *a* is a bootstrap factor obeying a normal distribution (mean, 0; standard deviation, 1). The food source locations are updated as follows:(7)$$x_{i}\left(t+1\right)=\left\{\begin{array}{c} x_{i}\left(t\right),\;f\left(x_{i}\left(t\right)\right)\leq f\left(v_{i}\left(t+1\right)\right)\\ v_{i}\left(t+1\right),\;f\left(x_{i}\left(t\right)\right)>f\left(v_{i}\left(t+1\right)\right) \end{array}\right.$$
where $x_{i}\left(t+1\right)$ denotes the position of the *i*th iteration of the food source, and f(·) denotes the fitness value of the function.

#### 2.1.3. Territorial Foraging

When rand≥0.5, hummingbirds forage regionally according to their current flight skill, searching for new food sources near their territory, with locations updated as follows:(8)$$v\left(t+1\right)=x_{i}\left(t\right)+b\times D\times x_{i}\left(t\right)$$
where *b* is a region factor that follows a normal distribution (mean, 0; standard deviation, 1).

#### 2.1.4. Migration Foraging

If t%(2×n)=0, migratory foraging is conducted to replace the food source with the worst existing nectar replenishment rate with other new food sources. The food source locations are updated as follows:(9)$$x_{wor}\left(t+1\right)=V_{L}+r\times \left(V_{U}-V_{L}\right)$$
where $x_{wor}$ indicates the location of the food source with the worst nectar replenishment rate in the population.

### 2.2. Support Vector Machine

Support vector machine (SVM) is a powerful and widely used machine learning algorithm for classification and regression problems. It offers numerous advantages, including the use of a small number of training samples, a short training time, and a simple structure [21]. The fundamental idea behind SVM is to find an optimal hyperplane that separates samples from different classes. In binary classification problems, the hyperplane can be viewed as an (n-1)-dimensional decision boundary, where n represents the number of features. The goal of the algorithm is to identify a hyperplane with the largest possible margin, thereby maximizing the separation between samples from different classes. The samples that are closest to the hyperplane are referred to as support vectors, as depicted in Figure 2.

The core idea of SVM is to address nonlinear problems by mapping the samples to a high-dimensional space and finding a linear hyperplane in that space. To tackle this problem, the concept of Lagrange multipliers is introduced to solve the quadratic optimization problem subject to inequality constraints. By solving this problem, a classification hyperplane can be obtained that maximizes the margin between different classes. Finally, the classification hyperplane is mapped back to the original input space, achieving the goal of nonlinear classification in the original input space. The classification function can be expressed as:(10)$$f\left(x\right)=\mathrm{sgn}\left(\sum_{i=1}^{N}\alpha_{i}y_{i}K\left(x_{i}\cdot x_{j}\right)+b\right)0<\alpha_{i}<C$$
where training sets T = $\{\left(x_{1},\;y_{1}\right),\;\left(x_{2},\;y_{2}\right),\;\ldots ,\;\left(x_{n},\;y_{n}\right)\},\;$$y_{i}\in Y=\left[-1,\;1\right]$, sgn represent the sign (+ or -) taken by the expression; $\alpha_{i}$ denotes the Lagrange multipliers; and *b* is the threshold determined after training the data. *C* represents the penalty factor, which controls the degree of punishment for misclassified samples. It balances the tradeoff between maximizing the margin and minimizing the classification errors. When the value of *C* is high, it indicates low tolerance for errors and may lead to overfitting issues. Conversely, when the value of *C* is low, it may result in underfitting problems. If the value of *C* is too large or too small, it can lead to a decrease in the model’s generalization ability. $K(x_{i}\cdot x_{j})$ represents the kernel function. In this study, the RBF kernel function is selected as the nonlinear mapping function for the SVM model. The expression for the RBF kernel function is as follows:(11)$$K\left(x_{i}\cdot x_{j}\right)=\exp(-||x_{i}-x_{j}{\left|\right|}^{2}/2g^{2})$$
where *g* is the kernel function parameter, and the value of *g* affects the speed and results of SVM model training and prediction.

## 3. Our Proposed IAHA Algorithm

In this section, we propose an improved version of the artificial hummingbird algorithm (AHA) called IAHA, which incorporates multiple enhancement strategies. We then compare IAHA with four basic algorithms, three single-stage improved AHA algorithms, and two existing improved artificial hummingbird algorithms. The comparison is conducted using eight benchmark test functions and simulation experiments and evaluated using the Wilcoxon rank-sum test.The experimental results demonstrate that IAHA outperforms the four basic algorithms, three single-stage improved AHA algorithms, and two existing improved artificial hummingbird algorithms. IAHA exhibits a faster convergence speed, stronger global optimization capability, and better overall algorithm performance.

### 3.1. Improvement Strategies

#### 3.1.1. Combining Tent Chaos Mapping with Backward Learning to Initialize Populations

The standard initialization strategy of AHA adopts a random initialization approach, which often leads to an uneven distribution of individuals in the population and insufficient population diversity. Previous research has shown that an effective population initialization strategy plays a crucial role in improving the optimization accuracy and convergence speed of the algorithm [22]. Chaotic mapping is frequently used to enhance the population initialization of intelligent optimization algorithms. Among them, Tent and logistic mapping are the most common chaotic models. However, Tent chaotic mapping exhibits better exploration capability and convergence speed compared to logistic chaotic mapping [23]. Therefore, in this study, we select Tent chaotic mapping as the foundation for population initialization. The formula for Tent chaotic mapping is as follows:(12)$$X_{\left(n+1\right)}=\left\{\begin{array}{c} 2\times X_{\left(n\right)},\;0\leq X_{\left(n\right)}\leq 0.5\\ 2\times \left(1-X_{\left(n\right)}\right),\;0.5<X_{\left(n\right)}\leq 1 \end{array}\right.$$
where $X_{n+1}\in \left[0,\;1\right]$. The initialized population ($X_{i,d}$) is obtained according to Tent chaos mapping. The inverse solution ($X_{{}_{i,d}}^{*}$) for the initialized population ($X_{i,\;d}$) is calculated as:(13)$$X_{{}_{i,d}}^{*}=Ub_{i,d}+Lb_{i,d}-X_{i,d}$$
where $Ub_{i,d}$ and $Lb_{i,d}$ are the maximum and minimum values of the *d* dimension in the *i*th individual corresponding to $X_{i,d}$, respectively. Then, the fitness values of each individual within $X_{i,d}$ and $X_{{}_{i,d}}^{*}$ are calculated, and the most outstanding individual is selected as the initial population individual after comparison using the following equation:(14)$$X_{i,d}=\left\{\begin{array}{c} X_{i,d},\;f_{i,d}<f_{i,d}^{*}\\ X_{{}_{i,d}}^{*},\;f_{i,d}\geq f_{i,d}^{*} \end{array}\right.$$
where $f_{i,d}$ and $f_{i,d}^{*}$ denote the fitness values of individuals in $X_{i,d}$ and $X_{{}_{i,d}}^{*}$, respectively.

#### 3.1.2. Levy Flight

Levy flight is a random flight simulated using a heavy-tailed distribution, where the individual takes small steps for a long time and occasionally takes large steps [24]. Introducing Levy flight into the algorithm ensures that the algorithm can perform a local search during small steps and disrupt the population positions during large steps, helping to escape local optima.In this study, the Levy flight strategy is incorporated into the food source updating process described in Equation (Equation 6). The improved formula is as follows:(15)$$v_{i}\left(t+1\right)=\left\{\begin{array}{c} v_{i1}\left(t+1\right),\;f\left(v_{i1}\left(t+1\right)\right)<f\left(v_{i2}\left(t+1\right)\right)\\ v_{i2}\left(t+1\right),\;f\left(v_{i1}\left(t+1\right)\right)\geq f\left(v_{i2}\left(t+1\right)\right) \end{array}\right.$$
where $v_{i1}\left(t+1\right)$ and $v_{i2}\left(t+1\right)$ are described as follows:(16)$$v_{i1}\left(t+1\right)=x_{i,tar}\left(t\right)+a\times D\times \left[x_{i}\left(t\right)-x_{i,tar}\left(t\right)\right]$$
(17)$$v_{i2}\left(t+1\right)=x_{i,tar}\left(t\right)+Levy\left(\lambda \right)\times \left[x_{i}\left(t\right)-x_{i,tar}\left(t\right)\right]$$

Equation (Equation 16) is the standard food source update method, and Equation (Equation 17) is the Levy flight-based food source update method. The fitness values of the two are compared, then selected based on merit. Equation (Equation 16), Levy(λ) is a random search path that satisfies:(18)$$Levy∼u=t^{-\lambda },\;1<\lambda \leq 3$$

Its step size (*s*), which obeys the Levy distribution, is calculated as follows:(19)$$s=\frac{\mu }{|v{|}^{1/\beta }}$$
where μ and *v* are normally distributed and are defined as:(20)$$\mu =N(0,\;\sigma_{\mu }^{2})$$
(21)$$v=N(0,\;\sigma_{v}^{2})$$
where
(22)$$\sigma_{\mu }=\frac{\left(1+\beta \right)\times \left(\sin\frac{\pi \beta }{2}\right)}{\frac{1+\beta }{2}\times \beta^{2}\times \frac{\beta -1}{2}}$$
(23)$$\sigma_{v}=1$$
where β is 1.5.

#### 3.1.3. Simplex Method for Optimization

The simplex method is a fast and simple polytope search algorithm [25] that involves reflecting, expanding, and searching for geometric transformations in the worst-performing individuals in a problem. This process generates better-quality individuals to replace them. The search points in the simplex method are illustrated in Figure 3.

The aim of the simplex method is to determine the location of the center point (*c*) on the basis of the best point (*g*), the second-best point (*b*), and the lowest point (*w*). The basic operation of the simplex method is as follows:

(1) Reflection operation:

The reflection point is r=c+α(c−w), where α is the reflection coefficient.

(2) Expansion operation:

When w>c, the extension point is calculated (e=c+β(r−c), where β is the extension factor). When e>g, the expansion point (*e*) is used instead of the closest point (*w*); otherwise, the reflection point (*r*) is used instead of the closest point (*w*).

(3) Compression operation

When g>r, the reflection direction is incorrect; then, the inward compression operation is performed (t=c+γ(w−c) where γ is the compression coefficient). When t>w, the compression point (*t*) is used instead of the closest point (*w*). When w<r<g, the outward compression operation (s=c−γ(w−c)) is performed. When s>w, the contraction point (*s*) is used instead of the closest point (*w*); otherwise, the reflection point (*r*) is used instead of the closest point (*w*).

In this paper, by using the simplex method, the algorithm optimizes the worst-quality individuals in the process of each iteration so as to improve the overall quality of the population, which can effectively solve the problem of the algorithm falling into a local optimum and improve the local search and optimality-seeking abilities of the algorithm.

#### 3.1.4. IAHA Execution Steps

After AHA is improved by the above three improvement strategies, the flow-specific implementation process of IAHA is shown in Algorithm 1.
**Algorithm 1** IAHA implementation steps.**Step 1:** Parameters such as population size, dimensionality, number of iterations, and upper and lower limits of the search space are set;
**Step 2:** The food source locations are initialized using a fused Tent chaos mapping and direction-learning strategy, and the corresponding fitness values are calculated. The access table is also initialized;
**Step 3:** Flight skills are randomly selected;
**Step 4:** The phase of guided foraging or area foraging begins based on the Levy flight strategy with a 50% probability of each of the two foraging behaviors. The visit table is updated after foraging behavior;
**Step 5:** When migratory foraging conditions are met, hummingbirds perform migratory foraging, randomly replacing the worst food source location. The access table is updated after foraging behavior;
**Step 6:** The location of poor food sources is optimized using the simplex method;
**Step 7:** The algorithm is terminated when the algorithm termination condition is met; otherwise, return to step 3.


The process of finding a solution in this study was conducted on an Intel Core i5 CPU with a clock speed of 2.50 GHz and 8 GB of memory. The experiments were performed in a Windows 10 64-bit testing environment using MATLAB R2022a software. The proposed IAHA algorithm was validated using eight classical test functions, as shown in Table 1. Among them, $f_{1}-f_{5}$ represent a unimodal function with a single unique extremum point within the interval that is used to test the algorithm’s development capability, and $f_{6}-f_{8}$ represent a multimodal function with multiple extremum points within the interval that can be used to test the algorithm’s ability to escape local optima. In the subsequent experiments, it is important to ensure that the algorithm parameters remain consistent across each experimental group. The population size is set to 30, the number of iterations is set to 500, and the function dimension is set to 30. For each function, 30 independent experiments are conducted, and the average value and standard deviation of the experiments are recorded to evaluate the algorithm’s performance.

### 3.2. Experimental Results

#### 3.2.1. Experimental Environment

#### 3.2.2. Comparison with Other Intelligent Optimization Algorithms

(1) Simulation results

To validate the effectiveness and feasibility of the algorithm proposed in this study, a comparative analysis is performed between the proposed IAHA algorithm and other algorithms, including PSO, WOA, GWO, and AHA. The analysis is conducted using the eight aforementioned test functions. The average values and standard deviations obtained by each algorithm are calculated and presented in Table 2. This comparison aims to assess the performance of the algorithms and provide insights into the superiority of the IAHA algorithm over the other methods.

Based on the analysis presented Table 2, it can be observed that in unimodal functions $f_{1}-f_{4}$, the IAHA algorithm achieves the theoretical optimum with a mean value, and the standard deviation is 0. In unimodal function $f_{5}$, the IAHA algorithm’s mean value is closer to the theoretical optimum, and the standard deviation is minimized. From this, it can be inferred that the IAHA algorithm exhibits a stronger exploration capability and stability during the optimization process.

For multimodal functions $f_{6}$ and $f_{8}$, the IAHA algorithm achieves an average value equal to the theoretical optimum, and the standard deviation is 0. In multimodal function $f_{7}$, the IAHA algorithm’s average value is closer to the theoretical optimum, and the standard deviation is 0, indicating consistent optimization results across multiple experiments. Therefore, it can be concluded that the IAHA algorithm exhibits strong optimization capabilities and can quickly escape local optima in multimodal test functions with multiple extreme points.

In conclusion, the IAHA algorithm demonstrates strong search capability and high stability in both unimodal and multimodal functions. It exhibits adaptability to different types of functions.

(2) Comparative analysis of convergence curves

Figure 4 depicts the convergence curves comparing IAHA with other basic intelligent optimization algorithms on various test functions. From Figure 4a–d,f,h, it is evident that the IAHA algorithm can rapidly converge to the theoretical optimum of the respective test functions. Even in the case of multimodal functions $f_{6}$ and $f_{8}$ in Figure 4f,h, where other algorithms also find the theoretical optimum, their convergence speed is significantly slower compared to that of the IAHA algorithm.

From Figure 4e,g, it is apparent that the IAHA algorithm does not find the theoretical optimum for test functions $f_{5}$ and $f_{7}$. However, in the case of the $f_{5}$ test function, although the theoretical optimum is not reached, the IAHA algorithm exhibits faster convergence and higher optimization accuracy compared to the other algorithms. In the $f_{7}$ test function, the IAHA algorithm achieves higher optimization accuracy and demonstrates significantly faster convergence, reaching a stable function value around 80 iterations, whereas the AHA algorithm, despite achieving the same accuracy, requires approximately 115 iterations to reach a stable function value.

In summary, compared to the other four basic algorithms, IAHA demonstrates excellent performance in terms of convergence speed and optimization accuracy. The convergence curves of IAHA consistently remain below those of the other algorithms, indicating higher optimization accuracy and faster convergence speed.

#### 3.2.3. Comparative Analysis with a Single-Improvement-Stage AHA Algorithm

(1) Simulation results

To verify the effectiveness of the different improvement strategies proposed in this paper, a comparison is conducted between IAHA, the original AHA algorithm, and the AHA algorithms with single-strategy improvements. These single-strategy improved algorithms, including TAHA, which incorporates the fusion of Tent chaotic mapping and a reverse learning strategy; LAHA, which solely incorporates the Levy flight strategy; and DAHA, which solely incorporates the simplex method. The comparison is conducted on the eight test functions presented in Table 1 to evaluate the effects of different improvement strategies and the performance of the multi-strategy improved artificial hummingbird algorithm. The experimental results are summarized in Table 3.

From the analysis in Table 3, it can be observed that in unimodal functions $f_{1}-f_{5}$, TAHA does not show a significant improvement in optimization accuracy compared to standard AHA. However, it consistently exhibits a smaller standard deviation, ensuring the stability of the algorithm. Therefore, by combining Tent chaotic mapping and the reverse learning strategy for population initialization, the algorithm can achieve a more stable optimization performance. In comparison to standard AHA, LAHA shows a significant improvement in optimization accuracy. This is due to the introduction of the Levy flight strategy in the algorithm, which enhances both local search accuracy and the ability to quickly escape local optima. While DAHA can find the theoretical optimum in the $f_{1}-f_{4}$ test functions, its optimization accuracy is not as good as that of other improvement strategies in the $f_{5}$ function. On the other hand, IAHA not only achieves the theoretical optimum in the $f_{1}-f_{4}$ functions but also maintains high optimization accuracy in the $f_{5}$ function. In multimodal functions $f_{6}-f_{8}$, all the improvement strategies yield the same optimization values. Additionally, in the $f_{6}$ and $f_{8}$ functions, all algorithms are able to find the theoretical optimum.

(2) Comparative analysis of convergence curves

Figure 5 shows the convergence curves of IAHA and the AHA algorithm with single-stage improvements in various test functions. Overall, TAHA consistently exhibits higher optimization accuracy than AHA and maintains algorithm stability. LAHA demonstrates significantly faster convergence compared to the other two single-stage improvement algorithms, enabling faster optimization. DAHA reaches the theoretical optimum of the corresponding function with fewer iterations, as shown in Figure 5a–d. IAHA combines the advantages of these three improvement strategies. It ensures better algorithm stability, faster convergence during optimization, and the ability to reach the theoretical optimum in fewer iterations. In conclusion, integrating these three improvement strategies into the AHA algorithm effectively enhances its capabilities.

#### 3.2.4. Comparative Analysis with Other Improved AHA Algorithms

(1) Simulation results

To further demonstrate the performance of the IAHA algorithm, we conduct comparative experiments with two recent improvement algorithms: CLAHA proposed by Wang et al. [26] and AOAHA proposed by Ramadan et al. [27]. The experimental results are presented in Table 4.

From Table 4, it can be observed that in the case of unimodal functions $f_{1}-f_{5}$, the IAHA algorithm proposed in this study outperforms the other two improvement algorithms in terms of optimization accuracy. However, in the case of multimodal functions $f_{6}-f_{8}$, the three improvement algorithms achieve similar optimization accuracy. Overall, the IAHA algorithm demonstrates superior optimization performance compared to the other two improvement algorithms.

(2) Comparative analysis of convergence curves

Figure 6 shows the convergence curves of IAHA and other improved versions of AHA in various test functions. Overall, it can be seen that the AOAHA algorithm only slightly improves the optimization accuracy and convergence speed compared to the original AHA algorithm. Although the CLAHA algorithm exhibits a noticeable improvement in convergence speed, the optimization accuracy still has room for improvement. On the other hand, the proposed IAHA algorithm demonstrates faster convergence speed in both unimodal functions ($f_{1}-f_{5}$) and multimodal functions ($f_{6}-f_{8}$). Overall, the IAHA algorithm’s convergence performance is superior to that of the other two improvement algorithms.

#### 3.2.5. Significance Analysis

To further elucidate the performance differences between the IAHA algorithm proposed in this study and other algorithms, such as PSO, WOA, GWO, AHA, TAHA, LAHA, BAHA, CLAHA, and AOAHA, the Wilcoxon rank-sum test is conducted. The experimental parameters are kept consistent with the previous settings, and the 30 independent runs are subjected to the rank-sum test. The results of the test are presented in Table 5.

a. $H_{0}$ indicates that there is no difference in the solution results of the comparison algorithms.

b. $H_{1}$ indicates a difference in the solution results of the comparison algorithms.

In the Wilcoxon rank-sum test, the significance level threshold is typically set to 0.05. When the calculated *p*-value is less than the significance level (p<0.05), we accept the alternative hypothesis ($H_{1}$) and conclude that there is a significant difference between IAHA and the compared algorithm. This suggests that IAHA exhibits a significantly different performance compared to the other algorithm being tested. On the other hand, when the calculated *p*-value is greater than or equal to the significance level (p≥0.05), we accept the null hypothesis ($H_{0}$) and conclude that there is no significant difference between IAHA and the compared algorithm in terms of the performance metric for the corresponding test function. This implies that IAHA and the compared algorithm have comparable performance on that specific test function. Similarly, when the calculated *p*-value is less than the significance level (p=NaN), we conclude that IAHA and the compared algorithm exhibit consistent solution accuracy on that test function without any significant difference. By utilizing a significance level threshold of 0.05, the Wilcoxon rank-sum test allows us to assess whether IAHA shows a significant difference in performance compared to the other algorithms being evaluated, thereby providing insights into the relative effectiveness of IAHA.

## 4. Our Proposed IAHA-SVM Coal Mine Environmental Safety Warning Model

In this section, we utilize the IAHA algorithm to optimize the penalty factor (*C*) and the kernel function parameter (*g*) of the SVM model, resulting in the development of the IAHA-SVM coal mine environmental safety level warning model. Based on the safety regulations of coal mines, the model classifies the safety levels into four categories. We then apply the model to classify coal mine safety data that we collected. By comparing the classification accuracy of the IAHA-SVM model with that of the AHA-SVM model, we aim to demonstrate the advantages of the improved algorithm integrated with the SVM model in this study.

The IAHA-SVM coal mine environmental safety level warning model is designed to provide accurate and reliable safety-level predictions for coal mine environments. By optimizing the SVM parameters using the IAHA algorithm, the model can effectively handle the complexities and variations in coal mine safety data, leading to improved classification performance. A comparison of the classification accuracy with that of the AHA-SVM model serves as an evaluation metric for the effectiveness of the IAHA-SVM model. If the IAHA-SVM model achieves higher accuracy in predicting the safety levels compared to the AHA-SVM model, it indicates that the integration of the IAHA algorithm has improved the performance of the SVM model for coal mine safety classification.This analysis provides evidence of the superiority of the IAHA-SVM model in accurately predicting coal mine safety levels. By leveraging the optimization capabilities of the IAHA algorithm, the IAHA-SVM model can better adapt to the characteristics of coal mine safety data, leading to enhanced classification accuracy and more reliable safety-level predictions.

### 4.1. Our Proposed IAHA-SVM Model

In the SVM model, the penalty factor (*C*) and the kernel function parameter (*g*) have a significant impact on the prediction accuracy of the model. Usually, these parameters are determined empirically and greatly affect the prediction results. Therefore, in this study, we employ the IAHA algorithm to optimize these parameters and select the values that lead to the best performance of the model. A flow chart of the proposed model is shown in Figure 7. The flow-specific implementation process of IAHA-SVM is shown in Algorithm 2.
**Algorithm 2** IAHA-SVM Execution Steps.**Step 1:** The collected coal mine safety-related data are divided into training and test sets and normalized;
**Step 2:** The SVM penalty-term coefficients (*C*); kernel function parameters (*g*); and IAHA-related parameters, including population size, maximum number of iterations, etc., are initialized;
**Step 3:** The food source locations are initialized using a fused Tent chaos mapping and direction-learning strategy, and the training-set samples are classified, with the SVM coal mine environmental safety classification accuracy as the individual fitness value;
**Step 4:** Flight skills are randomly selected;
**Step 5:** The phase of guided foraging or area foraging based on the Levy flight strategy begins, with a 50% probability of each of the two foraging behaviors. The visit table is updated after the foraging behavior;
**Step 6:** When migratory foraging conditions are met, hummingbirds perform migratory foraging, randomly replacing the worst food source location. The access table is updated after the foraging behavior;
**Step 7:** The location of poor food sources is optimized using the simplex method;
**Step 8:** The algorithm is terminated if the IAHA algorithm termination condition is met and the optimal *C* and *g* parameter values are output; otherwise, return to step 4;
**Step 9:** The IAHA-SVM coal mine environmental safety warning model is established.


### 4.2. Experimental Results

In this section of the simulation experiment, the population size is set to 20, and the maximum number of iterations is set to 50. The experimental data were obtained from a coal mine in Ningdong. Due to the scarcity of data in some extreme situations, in order to better meet the requirements of the experiment, part of the data are simulated based on real surface environmental data to form the dataset. This study classifies the coal mine environmental safety conditions into four levels according to the coal mine safety regulations, as shown in Table 6.

A D-level warning indicates a safe state where all parameters of the current environment are within a normal range, allowing for normal operation. A C-level warning indicates the occurrence of abnormal conditions, such as an increase in the concentration of flammable or explosive gases or leakage of toxic gases. The situation should be promptly investigated to eliminate hazards. A B-level warning triggers the underground alarm device when there are significant anomalies in the underground gas parameters. Personnel should be evacuated, and the situation should be investigated promptly. An A-level warning is the highest level of warning, indicating the occurrence of dangerous situations such as fire or massive leakage of toxic gases. Immediate evacuation and power cutoff should be carried out, and the situation should be investigated. Corresponding measures should be taken to resolve the issue when the parameters decrease to a B-level warning. In order to better ensure the safety of workers, this study determines the overall warning level based on the highest level of each parameter. For example, if the value of $O_{2}$ is 5%, which corresponds to the A level, but the ranges of other parameters are not in the A-level range, we consider this situation an overall A-level warning.

In this section, a total of 4120 data samples are selected for the simulation experiment. Among them, 4000 samples are used for training, with 1000 samples for each warning level. The remaining 120 samples are used for testing, with 30 samples for each warning level. The purpose of the experiment is to find the optimal penalty factor (*C*) and kernel function parameter (*g*) to achieve the best classification performance based on the accuracy of the training data levels. The parameter range is set from 0 to 1000. The experimental results are shown in Figure 8. From the figure, it can be observed that out of the 120 test data samples, only 2 data samples were misclassified, resulting in a classification accuracy of 98.3333%.

To demonstrate the improved performance of the proposed model, a comparative experiment is conducted by combining the basic AHA algorithm with SVM. The experimental results are presented in Table 7. From the table, it can be observed that the accuracy of the IAHA-SVM model is significantly higher than that of the AHA-SVM model. This indicates that the use of the IAHA algorithm to optimize the SVM model improves the precision of classification and reduces the impact of boundary values. As a result, the overall performance of the model is further enhanced.

## 5. Conclusions and Future Work

In this paper, we propose the IAHA-SVM coal mine environmental safety warning model. First, to address the limited global exploration capability and slow convergence speed of the artificial bee hummingbird algorithm (AHA), we employ a strategy that combines Tent chaotic mapping with reverse learning to initialize the population. In the foraging phase, the Levy flight strategy is introduced to enhance the search ability. Additionally, the Simplex method is incorporated at the end of each iteration to replace the worst solution. Comparative experiments are conducted to demonstrate the effectiveness of the IAHA algorithm. Next, we combine the improved IAHA algorithm with support vector machine (SVM) to optimize the SVM model’s penalty factor and kernel function parameters. The IAHA algorithm is used to search for the optimal parameters, and a coal mine environmental safety level warning model is established. The effectiveness of the proposed model is validated using a dataset generated from actual measurements in a coal mine in Ningdong Town, Ningxia. Compared to the SVM model optimized by the basic AHA algorithm, the IAHA-SVM model shows superior performance.

Integrating intelligent optimization algorithms with real-world application problems can enhance work efficiency. The IAHA-SVM model proposed in this paper can also be applied to classification problems in various other domains. In our future work, we will continue to explore the application of intelligent optimization algorithms in different aspects of coal mine safety. Some potential directions include optimizing the deployment of wireless sensors underground, addressing underground positioning problems, and solving three-dimensional path planning for underground drones.

## Figures and Tables

**Figure 1 sensors-23-06614-f001:**
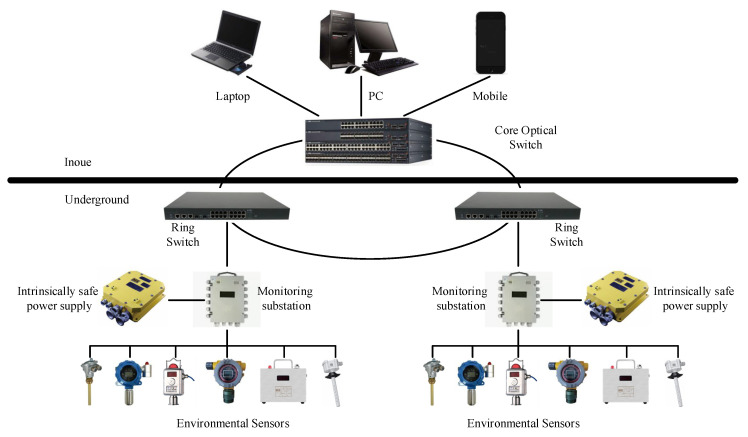
Framework of underground coal mine environmental monitoring system.

**Figure 2 sensors-23-06614-f002:**
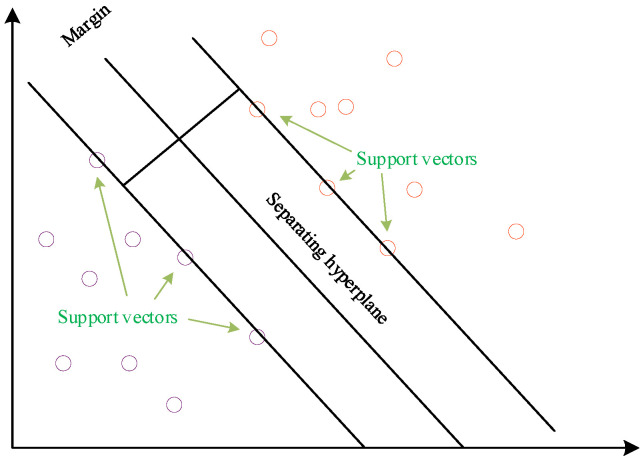
Optimal partition plane.

**Figure 3 sensors-23-06614-f003:**
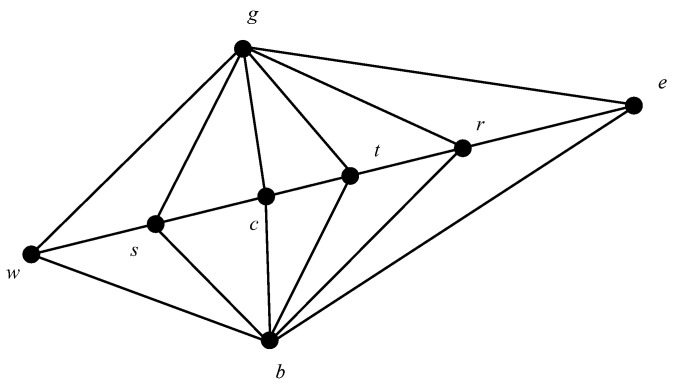
Search points of the simplex method.

**Figure 4 sensors-23-06614-f004:**
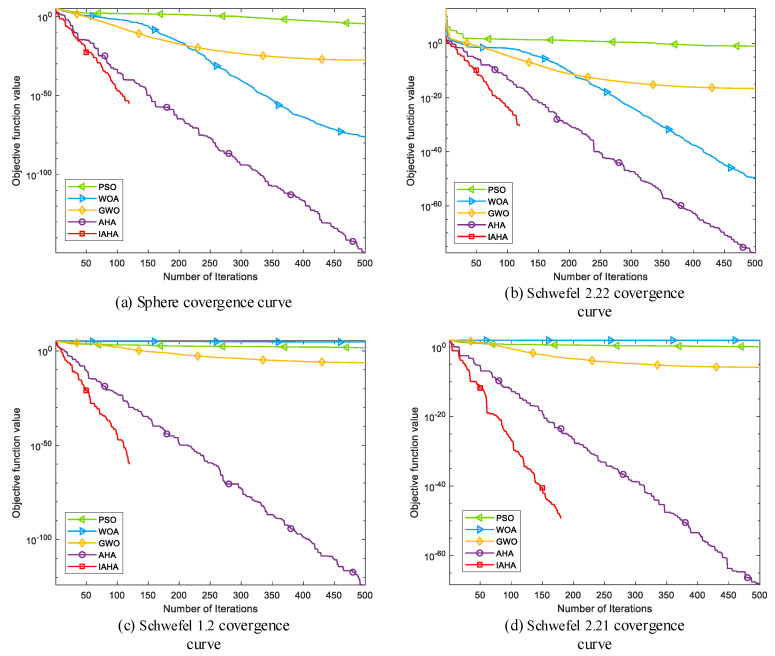
Comparison of IAHA and basic algorithm convergence curves.

**Figure 5 sensors-23-06614-f005:**
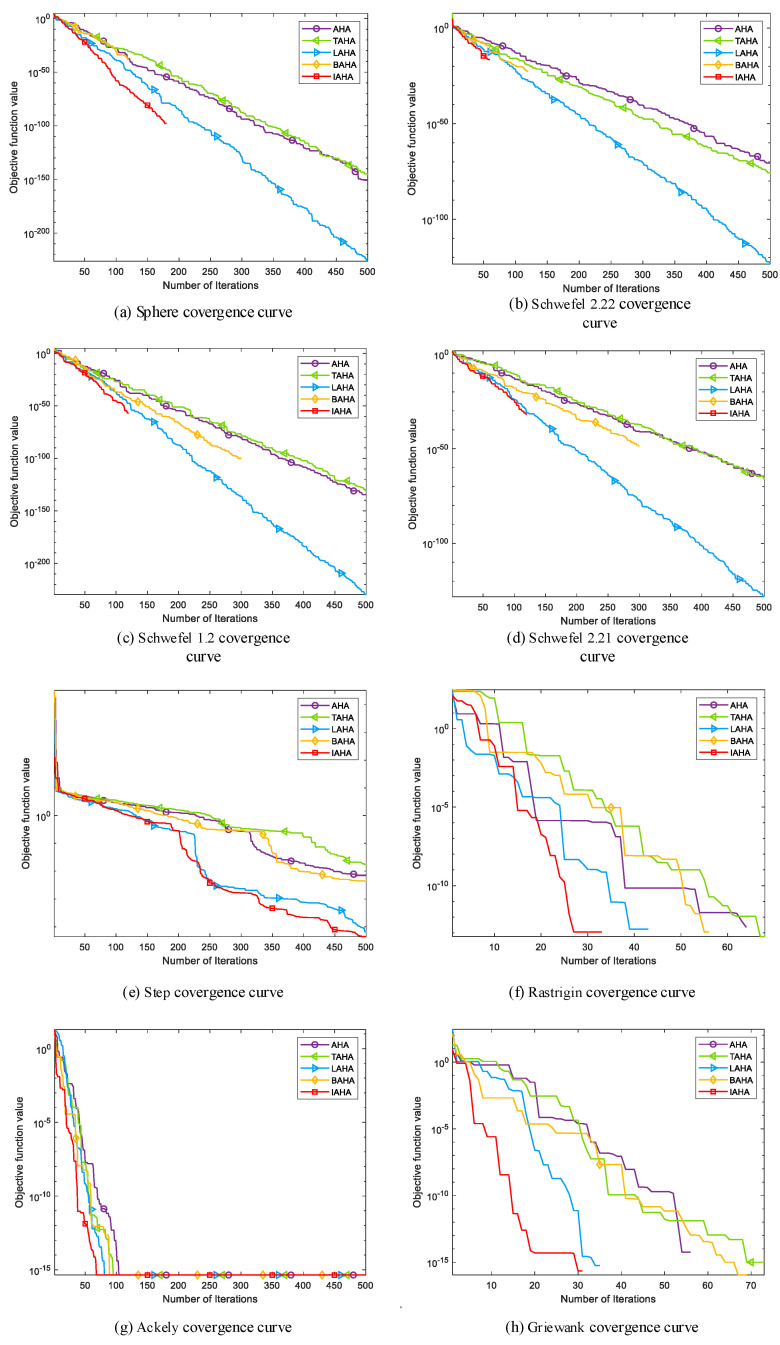
Comparison of IAHA and single-improvement-phase AHA convergence curves.

**Figure 6 sensors-23-06614-f006:**
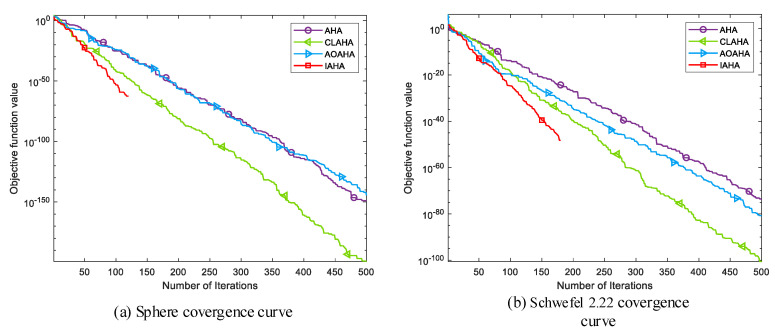
Comparison of IAHA with other improved AHA convergence curves.

**Figure 7 sensors-23-06614-f007:**
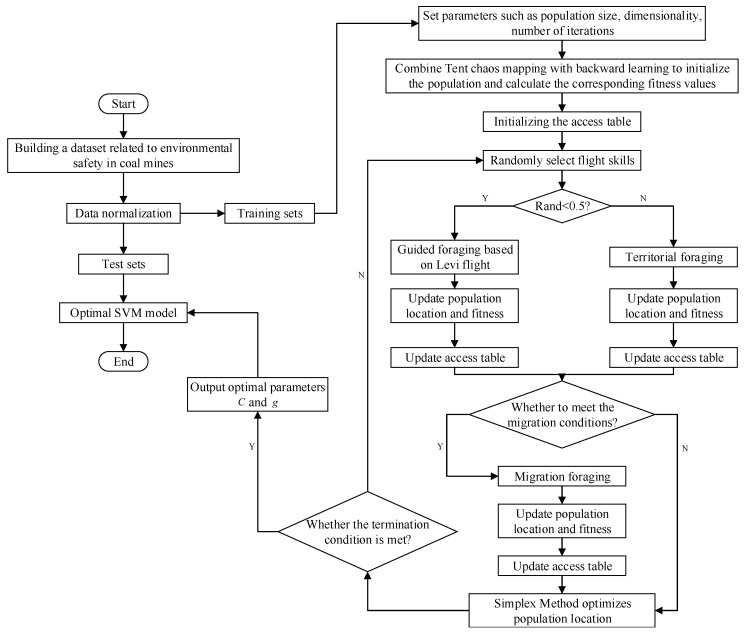
IAHA-SVM flow chart.

**Figure 8 sensors-23-06614-f008:**
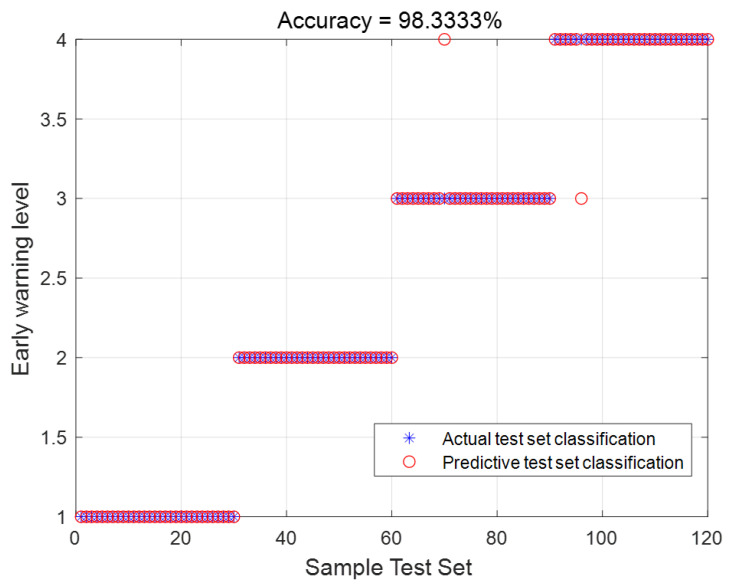
Actual and predicted classification plots for the test set.

**Table 1 sensors-23-06614-t001:** Test function.

Number	Function Name	Definition	Function Value	Optimal Value
(1)	Sphere	$f\left(x\right)=\sum_{i=1}^{n}x_{i}^{2}$	[−100 , 100]	0
(2)	Schwefel 2.22	$f\left(x\right)=\sum_{i=1}^{n}\left|x_{i}\right|+\prod_{i=1}^{n}\left|x_{i}\right|$	[−10, 10]	0
(3)	Schwefel 1.2	$f\left(x\right)=\sum_{i=1}^{n}{\left(\sum_{j=1}^{i}x_{j}\right)}^{2}$	[−100, 100]	0
(4)	Schwefel 2.21	$f_{4}\left(x\right)=\max_{i=1}^{n}\left|x_{i}\right|$	[−100, 100]	0
(5)	Step	$f\left(x\right)=\sum_{i=1}^{n}{\left(\left|x_{i}+0.5\right|\right)}^{2}$	[−100, 100]	0
(6)	Rastrigin	$f\left(x\right)=10n+\sum_{i=1}^{n}\left[x_{i}^{2}-10\cos\left(2\pi x_{i}\right)\right]$	[−5.12, 5.12]	0
(7)	Ackley	$f\left(x\right)=20+e-20\exp\left(-0.2\sqrt{\sum_{i=1}^{n}\frac{x_{i}^{2}}{n}}\right)-\exp\left(\sqrt{\sum_{i=1}^{n}\frac{\cos(2\pi x_{i})}{n}}\right)$	[−32, 32]	0
(8)	Griewank	$f\left(x\right)=\sum_{i=1}^{n}\frac{x_{i}^{2}}{4000}-\prod_{i=1}^{n}\cos(\frac{x_{i}}{\sqrt{i}})+1$	[−600, 600]	0

**Table 2 sensors-23-06614-t002:** Comparison of IAHA and basic algorithm experimental results.

Function	Evaluation	PSO	WOA	GWO	AHA	IAHA
Sphere	Mean	$9.15\times 10^{-5}$	$1.97\times 10^{-71}$	$2.22\times 10^{-27}$	$1.49\times 10^{-140}$	0
	SD	$9.38\times 10^{-5}$	$5.07\times 10^{-72}$	$4.72\times 10^{-27}$	$5.72\times 10^{-140}$	0
Schwefel 2.22	Mean	$3.71\times 10^{-2}$	$2.78\times 10^{-52}$	$5.98\times 10^{-17}$	$1.59\times 10^{-71}$	0
	SD	$2.93\times 10^{-2}$	$5.00\times 10^{-52}$	$3.49\times 10^{-17}$	$6.15\times 10^{-71}$	0
Schwefel 1.2	Mean	$1.11\times 10^{2}$	$4.45\times 10^{4}$	$1.23\times 10^{-5}$	$4.78\times 10^{-135}$	0
	SD	3.42×10	$1.47\times 10^{4}$	$3.29\times 10^{-5}$	$1.59\times 10^{-134}$	0
Schwefel 2.21	Mean	1.01	4.16×10	$4.79\times 10^{-7}$	$7.36\times 10^{-66}$	0
	SD	$2.25\times 10^{-1}$	2.36×10	$4.72\times 10^{-7}$	$2.72\times 10^{-65}$	0
Step	Mean	$1.97\times 10^{-4}$	$4.49\times 10^{-1}$	$7.48\times 10^{-1}$	$4.66\times 10^{-2}$	$7.94\times 10^{-5}$
	SD	$1.22\times 10^{-4}$	$2.50\times 10^{-1}$	$4.64\times 10^{-1}$	$9.87\times 10^{-2}$	$4.15\times 10^{-5}$
Rastrigin	Mean	5.65×10	$7.58\times 10^{-15}$	2.53	0	0
	SD	1.64×10	$2.94\times 10^{-14}$	$3.69\times 10^{-1}$	0	0
Ackley	Mean	$3.17\times 10^{-1}$	$3.76\times 10^{-15}$	$1.04\times 10^{-13}$	$4.44\times 10^{-16}$	$4.44\times 10^{-16}$
	SD	$5.23\times 10^{-1}$	$2.50\times 10^{-15}$	$1.85\times 10^{-14}$	0	0
Griewank	Mean	$7.73\times 10^{-3}$	$1.97\times 10^{-2}$	$2.01\times 10^{-3}$	0	0
	SD	$7.08\times 10^{-3}$	$5.22\times 10^{-2}$	$5.62\times 10^{-3}$	0	0

Mean indicates mean, and SD indicates standard deviation.

**Table 3 sensors-23-06614-t003:** Comparison of IAHA and single-improvement-phase experimental results.

Function	Evaluation	AHA	TAHA	LAHA	DAHA	IAHA
Sphere	Mean	$7.23\times 10^{-140}$	$3.41\times 10^{-141}$	$6.57\times 10^{-221}$	0	0
	SD	$1.86\times 10^{-139}$	$1.32\times 10^{-140}$	0	0	0
Schwefel 2.22	Mean	$4.94\times 10^{-72}$	$2.58\times 10^{-76}$	$1.65\times 10^{-113}$	0	0
	SD	$1.88\times 10^{-71}$	$4.24\times 10^{-76}$	$6.37\times 10^{-113}$	0	0
Schwefel 1.2	Mean	$1.34\times 10^{-122}$	$6.48\times 10^{-132}$	$1.20\times 10^{-216}$	0	0
	SD	$5.17\times 10^{-122}$	$1.98\times 10^{-131}$	0	0	0
Schwefel 2.21	Mean	$3.38\times 10^{-65}$	$2.09\times 10^{-66}$	$2.02\times 10^{-119}$	0	0
	SD	$8.05\times 10^{-65}$	$3.64\times 10^{-66}$	$7.75\times 10^{-119}$	0	0
Step	Mean	$4.94\times 10^{-2}$	$1.06\times 10^{-2}$	$1.14\times 10^{-4}$	$5.74\times 10^{-2}$	$6.39\times 10^{-5}$
	SD	$9.10\times 10^{-2}$	$5.99\times 10^{-3}$	$9.68\times 10^{-5}$	$1.05\times 10^{-1}$	$4.73\times 10^{-5}$
Rastrigin	Mean	0	0	0	0	0
	SD	0	0	0	0	0
Ackley	Mean	$4.44\times 10^{-16}$	$4.44\times 10^{-16}$	$4.44\times 10^{-16}$	$4.44\times 10^{-16}$	$4.44\times 10^{-16}$
	SD	0	0	0	0	0
Griewank	Mean	0	0	0	0	0
	SD	0	0	0	0	0

**Table 4 sensors-23-06614-t004:** Comparison of IAHA and other improved AHA experimental results.

Function	Evaluation	AHA	CLAHA	AOAHA	IAHA
Sphere	Mean	$2.50\times 10^{-135}$	$9.16\times 10^{-197}$	$5.19\times 10^{-150}$	0
	SD	$7.59\times 10^{-135}$	0	$1.45\times 10^{-149}$	0
Schwefel 2.22	Mean	$1.14\times 10^{-73}$	$6.51\times 10^{-102}$	$1.44\times 10^{-75}$	0
	SD	$3.63\times 10^{-73}$	$1.31\times 10^{-101}$	$4.11\times 10^{-75}$	0
Schwefel 1.2	Mean	$3.42\times 10^{-134}$	$2.00\times 10^{-186}$	$1.82\times 10^{-135}$	0
	SD	$1.32\times 10^{-133}$	0	$3.56\times 10^{-135}$	0
Schwefel 2.21	Mean	$6.91\times 10^{-65}$	$3.04\times 10^{-107}$	$1.13\times 10^{-64}$	0
	SD	$1.74\times 10^{-64}$	$1.10\times 10^{-106}$	$3.26\times 10^{-64}$	0
Step	Mean	$4.69\times 10^{-2}$	$5.85\times 10^{-1}$	$3.19\times 10^{-2}$	$1.61\times 10^{-4}$
	SD	$1.02\times 10^{-1}$	$3.17\times 10^{-1}$	$6.74\times 10^{-2}$	$1.65\times 10^{-4}$
Rastrigin	Mean	0	0	0	0
	SD	0	0	0	0
Ackley	Mean	$4.44\times 10^{-16}$	$4.44\times 10^{-16}$	$4.44\times 10^{-16}$	$4.44\times 10^{-16}$
	SD	0	0	0	0
Griewank	Mean	0	0	0	0
	SD	0	0	0	0

**Table 5 sensors-23-06614-t005:** Comparison of IAHA and single-improvement-phase experimental results.

Function	PSO	WOA	GWO	AHA	TAHA	LAHA	DAHA	CLAHA	AOAHA
Sphere	0	0	0	0	0	0	NaN	0	0
Schwefel 2.22	0	0	0	0	0	0	NaN	0	0
Schwefel 1.2	0	0	0	0	0	0	NaN	0	0
Schwefel 2.21	0	0	0	0	0	0	NaN	0	0
Step	0	0	0	0	0	0.24	0	0	0
Rastrigin	0	0.35	0	NaN	NaN	NaN	NaN	NaN	NaN
Ackley	0	0	0	NaN	NaN	NaN	NaN	NaN	NaN
Griewank	0	0.09	0.09	NaN	NaN	NaN	NaN	NaN	NaN

The values in the table are significance levels (*p*) when p<0.05, as denoted by 0.

**Table 6 sensors-23-06614-t006:** Comparison of IAHA and single-improvement-phase experimental results.

Warning Level	$T^{\circ }C$	$O_{2}\%$	$CO_{2}\%$	CO%	$NH_{3}\%$	$H_{2}S\%$	$SO_{2}\%$	$CH_{4}\%$	Symbol
A	≥80	≤8.4	≥0.41	≥0.0024	≥0.0041	≥0.00067	≥0.001	≥5	4
B	30–79.9	8.5–15.4	0.16–0.40	0.0016–0.0023	0.0021–0.004	0.00045–0.00066	0.0005-0.0099	1.6–4.9	3
C	23–29.9	15.5–19.4	0.06–0.15	0.0006–0.0015	0.0006–0.0020	0.00023–0.00044	0.0002–0.00049	0.59–1.59	2
D	16–22.9	19.5–23.5	0–0.05	0–0.0005	0–0.0005	0–0.00022	0–0.00019	0-0.58	1

**Table 7 sensors-23-06614-t007:** Experimental results.

Model	Penalty Factor (*C*)	Parameter (*g*)	Accuracy Rate %
AHA-SVM	548.744	62.098	94.1667
IAHA-SVM	123.979	0.891	98.3333

## Data Availability

The corresponding author can provide data supporting the findings of this study upon request. Due to ethical or privacy concerns, the data are not publicly available.

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
