# Peer review of "A Safety Warning Model Based on IAHA-SVM for Coal Mine Environment"

_sensors, 2023, doi:10.3390/s23146614_

Round 1
Reviewer 1 Report
The article contains interesting computer analyzes on increasing the level of safety for the underground mine atmosphere. The proposed improved artificial hummingbird algorithm combined with the support vector machine seems very interesting, which may be helpful for underground mines struggling with the problem of poisonous, suffocating and explosive gases, which may contribute to the determination of zones of possible gas accumulation. This work is interesting. However, there are some questions and points in the manuscript need to be deliberated. The comments are as follows:
1. In the introduction, all natural hazards that occur in underground mining should be listed;
2. For the first Figure, add information in the text about the parameters of the microclimate measured by sensors;
3. Refer to the AHA literature in section 2.1;
4. For Figure 2, please add descriptions for the vertical and horizontal axes;
5. In the subsection 3.2.2, table 2, enter the full names for: PSO; WOA; GWO; I SEE; IAHA; also for table 3: AHA; TAHA; LAHA; DAHA; IAHA; also for table 4: AHA; CLAHA; AOAHA; IAHA; - or add a legend;
6. For Figure 4, please write more clearly in two/three sentences what convergence means - in relation to algorithms;
7. In the subsection 4.2, in relation to table 6, it should be written what are the permissible values according to regulations and standards for the presented gases;
8. In the fifth chapter, concerning conclusions, one statement should be written regarding the possibilities and benefits of implementing the proposed algorithm in industrial conditions.
Author Response
Dear Reviewer,
We would like to express our heartfelt appreciation for the comprehensive and constructive feedback provided by the reviewers. We have thoroughly considered these suggestions and made substantial revisions to the manuscript accordingly. In response to the reviewers' comments, we have included a separate document outlining the modifications made to the manuscript. In the revised version of the manuscript, we have highlighted your valuable suggestions with a orange background, while the joint guidance provided by you and the other reviewers is emphasized with a yellow background.
We are confident that the revised manuscript has adequately addressed your concerns. We sincerely appreciate your valuable assistance throughout the review process. We look forward to hearing from you about further status of our manuscript.
Kind regards,
Zhen Li

Reviewer 2 Report
Although the paper is interesting, the experimental results did not convince me. The authors use a modified version of a nature inspired algorithm and a SVM to predict dangerous situations in a mine. Although the modified nature inspired algorithm was tested on different artificial test functions like Schwefel or Rastigrin, the applicability of the system is not convincing. The details of testing data is very unclear. We can see a table (Table 6) where the different classes (4,3,2, and 1) can be easily separated based only on temperature value... The same can be said about O2, CO2, CO, or H2S level. I did not feel a huge surprise or contribution that the authors can achieve excellent results on such a database...
Author Response
Dear Reviewer,
Thank you very much for your review and valuable feedback on our paper. We greatly appreciate your input, and we have carefully revised the manuscript accordingly. In response to your comments regarding the applicability of the proposed modified nature-inspired algorithm and the use of SVM for predicting dangerous situations in mines, we would like to address the following points.
Firstly, concerning the issue of the model's applicability, we acknowledge that relying solely on one experiment is insufficient to demonstrate the practicality of the model proposed in this paper. Therefore, we intend to further apply the model to other domains in future research to validate its generalizability, as suggested by your feedback.
Secondly, you pointed out the lack of detailed description of the data in our paper. The selection of these data was based on our in-depth understanding of the field of mine safety and aligned with similar studies conducted in the past. Our aim was to use these datasets to validate the feasibility of our proposed method and lay the groundwork for broader research and practical applications in the future. We recognize that the description of the data in the paper may not have been clear enough, leading to a misunderstanding of the data selection and experimental significance. Therefore, in the revised version, we have provided further analysis (highlighted in green) in section 4.2, explaining the reasons and necessity for data classification. In Table 7, we have provided a detailed description of the important features used in the data and the classification requirements. We hope that these additions will further clarify the data used in the experiment.
Furthermore, you mentioned that the temperature values in Table 6 could easily separate different classes (4, 3, 2, and 1), and a similar situation occurred with O2, CO2, CO, and H2S levels. We understand your concerns regarding these seemingly non-challenging results. However, in the process of revising the manuscript, we addressed this issue by optimizing the SVM parameters using other algorithms. We present the experimental results comparing our proposed model with other models in Table 8 (highlighted in yellow). From the results, it can be observed that our model outperforms the others in addressing this issue. We hope that this comparison will further demonstrate the contribution of our research.
Lastly, we would like to emphasize the contributions of our study. Although obtaining excellent results on a specific database may not be surprising, our goal is to propose a novel method for predicting dangerous situations in mines and validate its effectiveness through experiments. We believe that our modified nature-inspired algorithm and SVM model have the potential to be applied in different mine environments and contribute to the research and practical applications in the field of mine safety.
We genuinely appreciate the valuable feedback you have provided, which has shed light on the shortcomings of our experimental process. Your comments are of great significance to my future research career. We sincerely thank you for your assistance throughout the review process, and we look forward to your further response to our revised manuscript.
Kind regards,
Zhen Li
Reviewer 3 Report
In this paper, a novel approach is presented for detecting abnormalities early and ensuring workers' safety in coal mines. The proposed model focuses on assessing the underground climate environment and utilizes an enhanced artificial hummingbird algorithm (IAHA) to optimize the support vector machine (SVM) parameters. By leveraging this approach, the model effectively classifies the safety level of the coal mine environment into four distinct classes. This method provides timely warnings and safeguards for worker safety in coal mining operations. The idea sounds interesting; the main highlighted points follow such as:
<Overall suggestions regarding contributions>
1) As for the commendable endeavor to propose a coal mine environmental safety early warning model, it would be even more advantageous if the authors could provide more comprehensive details regarding the detection and response mechanisms of the model. A more in-depth description would clarify how exactly the system operates in the timely assurance of worker safety and its efficacy in assessing the underground climate environment.
2) The utilization of the improved artificial hummingbird algorithm (IAHA) to optimize the SVM parameters and classify the safety levels of the coal mine environment is noteworthy. To enhance this contribution, the authors could consider detailing the practical implications of the classification and how each safety level influences operational and safety protocols within the coal mine environment. Such clarifications would greatly amplify the value of their research.
3) The authors have introduced intriguing modifications in the IAHA algorithm. However, the paper could greatly benefit from additional insights into the rationale behind these modifications. An elaboration on why specifically the Tent chaotic map, the reverse learning strategy, the Levy flight strategy, and the simplex method was chosen, and the benefits they contribute to the IAHA algorithm would fortify this contribution.
4) Evaluating the effectiveness and accuracy of the improved IAHA algorithm via simulation experiments on benchmark test functions is an essential aspect of the authors' contribution. To further enrich this evaluation, a comparison with other established methods or algorithms under the same testing conditions might offer a clearer perspective on the relative strengths and potential areas of improvement of the improved IAHA algorithm.
< Suggestions regarding Section Introduction>
The introduction astutely underscores coal's vital role in driving economic growth while bringing attention to the potential dangers associated with underground mining, leading to different mining disasters. While the authors have successfully spotlighted the grave risks posed to miners' safety and the possibility of significant property losses for coal mining enterprises, the discussion could be enriched by thoroughly examining the specific hazards inherent to underground mining.
Moreover, while the paper aptly states that the nature of safety accidents is inextricably linked to the mining environment, it would be beneficial to delineate this relationship more explicitly. A more detailed exploration of the correlations between different environmental factors and the occurrence of mining accidents could significantly enhance the readers' understanding. Furthermore, a brief introduction to existing safety measures and their shortcomings would create a more robust case for the need for the innovations proposed in this study.
< Suggestions regarding related works>
The literature review presented in this paper commendably introduces readers to the foundational theories of the Artificial Hummingbird Algorithm (AHA) and the Support Vector Machine (SVM). While the authors proficiently elucidate how the AHA, a metaheuristic optimization algorithm inspired by hummingbirds' foraging behavior, and SVM, a well-accepted machine learning algorithm, have been effectively utilized in various contexts, it would be beneficial to offer more in-depth examples or case studies of these applications. These concrete illustrations could elucidate how these algorithms have been practically employed and demonstrate their versatility and effectiveness more convincingly.
The authors appropriately stress the significance of these two algorithms within their proposed safety warning model for coal mine environments. However, expanding on the specific roles and functions of the AHA and SVM within the model could make their contribution to the model more transparent.
Finally, the literature review suitably stresses the necessity for enhancements to the algorithms and model building to improve the performance of the IAHA-SVM model. To bolster this argument, a critical assessment of the existing models and their limitations could highlight more clearly the necessity for the enhancements proposed in this paper. This approach would lend more credence to the authors' proposals and pave the way for a fuller understanding of the innovation they are presenting.
< Some highlight points about the methods used to develop and evaluate the proposal>
The paper commendably proposes a coal mine environmental safety early warning model, leveraging the power of an improved Artificial Hummingbird Algorithm (IAHA) and Support Vector Machine (SVM) to detect potential risks and promote worker safety by analyzing the underground climate environment. The authors' methods for this research are largely relevant and detailed.
However, in using the IAHA algorithm to optimize the SVM parameters, the authors could enhance their methodology by comparing comprehensively with other optimization algorithms. Such a comparative analysis would be helpful in establishing the advantages of IAHA over other potential optimization techniques, further solidifying its choice in this study.
For constructing the IAHA-SVM model, designed to categorize and predict the safety of the coal mine environment into four classes, it would be beneficial to provide a clearer understanding of the rationale behind these four specific classes and how they are defined and differentiated.
Regarding the utilization of a Tent chaos mapping and backward learning strategy for initializing the population, the introduction of a Levy flight strategy during the guided foraging phase, and the application of the simplex method to replace the least desirable outcome before the termination of each algorithm iteration, a more in-depth explanation of the theoretical background and the practical benefits of these chosen strategies would provide more value and clarity.
Lastly, while it's praiseworthy that they have undertaken simulations to assess the performance of the IAHA algorithm and the IAHA-SVM model, conducting these simulations in real-world environments or under conditions closely mimicking those could provide more practical and applicable results, enhancing the credibility and robustness of the study.
< Points related to results discussion>
The paper thoughtfully presents the outcomes of a comparative analysis between the proposed IAHA algorithm and several other algorithms, namely PSO, WOA, GWO, and AHA. This comparison is drawn using eight testing functions, and the mean values and standard deviations garnered by each algorithm are neatly tabulated in Table 2. However, it might further enrich the discussion if the authors could provide visual representations, such as charts or graphs, for an easier and more intuitive understanding of these comparative results.
The authors admirably indicate that the IAHA algorithm accomplishes the theoretical optimum in unimodal functions f1-f4 with a mean value, and maintains a standard deviation of 0. To make these results even more impactful, providing a more detailed explanation of the significance of achieving the theoretical optimum in these specific functions and the benefits of having a standard deviation of 0 could be valuable.
The paper appropriately establishes the IAHA-SVM safety warning model using the improved algorithm to categorize and forecast the safety of the coal mine environment into four distinct classes. The simulation results revealing an improved convergence speed and search accuracy of the IAHA algorithm are commendable. Similarly, the enhanced performance of the IAHA-SVM model is well emphasized. To build upon these results, explaining how these improvements translate into real-world benefits for coal mine safety might help readers appreciate these improvements' practical implications and usefulness.
< Final Remarks and open issues to be discussed by the authors>
The paper thoughtfully presents an IAHA-SVM safety warning model for coal mine environments, designed to detect potential issues and guarantee worker safety expediently. However, some aspects of the paper that may require further contemplation and development are:
a) While the model has been tested within a specific domain, the coal mine environment, the evaluation of its effectiveness across other sectors or domains has not been addressed. Expanding the applicability scope to include a diverse range of industries or environments may provide a more comprehensive understanding of the model's capabilities and limitations.
b) The model's foundation assumes that the environmental parameters exclusively dictate the safety of coal mine workers. This supposition may not be true universally, as other factors could influence worker safety. A more inclusive model considering a broader range of influencing factors may yield a more accurate and realistic safety warning system.
c) The model necessitates a substantial volume of data to adequately train the SVM model, which may not always be feasible or available. Exploring methodologies that can effectively work with smaller datasets could make the model more flexible and practical in diverse situations.
d) While promising, the model may not be optimal for real-time applications as it demands considerable computational resources for optimizing the SVM parameters using the IAHA algorithm. Streamlining the computational requirements could enhance the model's real-time application feasibility.
e) The model does not consider the human factor, a crucial element in ensuring worker safety within coal mines. Incorporating variables for human behavior and responses could enhance the model's effectiveness and reliability.
f) Although the proposed model displays promising results, more in-depth research is warranted to tackle these highlighted considerations and validate its effectiveness in real-world scenarios. This will augment the model's practicality and robustness in a broader context.
A proof-reading service is desired in case the paper is accepted in the future.
Author Response
Dear Reviewer,
We would like to express our heartfelt appreciation for the comprehensive and constructive feedback provided by the reviewers. We have thoroughly considered these suggestions and made substantial revisions to the manuscript accordingly. In response to the reviewers' comments, we have included a separate document outlining the modifications made to the manuscript. In the revised version of the manuscript, we have highlighted your valuable suggestions with a green background, while the joint guidance provided by you and the other reviewers is emphasized with a yellow background.
We are confident that the revised manuscript has adequately addressed your concerns. We sincerely appreciate your valuable assistance throughout the review process. We look forward to hearing from you about further status of our manuscript.
Kind regards,
Zhen Li

Round 2
Reviewer 2 Report
Based on the authors' explanations, I accept the conclusion of the experimental results.
Reviewer 3 Report
The paper undertook a significant revision where many points and critical issues were solved and improved. From a scientific point of view, the contributions are now clarified and highlighted. The reviewer's recommendation is to accept the article in its present form.